# Synchronous Colorectal Cancer: Improving Accuracy of Detection and Analyzing Molecular Heterogeneity—The Main Keys for Optimal Approach

**DOI:** 10.3390/diagnostics11020314

**Published:** 2021-02-15

**Authors:** Patricia Simu, Ioan Jung, Laura Banias, Zsolt Kovacs, Zsolt Zoltan Fulop, Tivadar Bara, Iunius Simu, Simona Gurzu

**Affiliations:** 1Department of Radiology and Imaging, ‘George Emil Palade’ University of Medicine, Pharmacy, Sciences and Technology, 530149 Targu Mures, Romania; simupatricia@gmail.com (P.S.); iunius_simu@yahoo.com (I.S.); 2Department of Pathology, ‘George Emil Palade’ University of Medicine, Pharmacy, Sciences and Technology, 530149 Targu Mures, Romania; jungjanos@studium.ro (I.J.); laurabanias@gmail.com (L.B.); 3Department of Pathology, Clinical County Emergency Hospital, 530150 Targu Mures, Romania; kovacska_zsoltkovacs@yahoo.com; 4Department of Biochemistry, ‘George Emil Palade’ University of Medicine, Pharmacy, Sciences and Technology, 530149 Targu Mures, Romania; 5Department of Surgery, ‘George Emil Palade’ University of Medicine, Pharmacy, Sciences and Technology, 530149 Targu Mures, Romania; zsolt_fulop15@yahoo.com (Z.Z.F.); barativadar@yahoo.com (T.B.); 6Research Center (CCAMF), ‘George Emil Palade’ University of Medicine, Pharmacy, Sciences and Technology, 540139 Targu Mures, Romania

**Keywords:** synchronous tumors, colorectal cancer, microsatellite status, KRAS, BRAF, E-cadherin

## Abstract

Background: In patients with synchronous colorectal cancer (SCRC), understanding the underlying molecular behavior of such cases is mandatory for designing individualized therapy. The aim of this paper is to highlight the importance of transdisciplinary evaluation of the pre- and post-operative assessment of patients with SCRCs, from imaging to molecular investigations. Methods: Six patients with SCRCs presented with two carcinomas each. In addition to the microsatellite status (MSS), the epithelial mesenchymal transition was checked in each tumor using the biomarkers β-catenin and E-cadherin, same as KRAS and BRAF mutations. Results: In two of the patients, the second tumor was missed at endoscopy, but diagnosed by a subsequent computed-tomography-scan (CT-scan). From the six patients, a total of 11 adenocarcinomas (ADKs) and one squamous cell carcinoma (SCC) were analyzed. All the examined carcinomas were BRAF-wildtype microsatellite stable tumors with an epithelial histological subtype. In two of the six cases, KRAS gene status showed discordance between the two synchronous tumors, with mutations in the index tumors and wildtype status in the companion ones. Conclusions: Preoperative CT-scans can be useful for detection of synchronous tumors which may be missed by colonoscopy. Where synchronous tumors are identified, therapy should be based on the molecular profile of the indexed tumors.

## 1. Introduction

Colorectal cancer (CRC) is a heterogenous tumor and the third most common type of malignancy worldwide [1,2,3]. It accounts for 10% of new cancer diagnoses each year, and it is the second leading cause of cancer deaths in males and the third leading cause of cancer death in females [2,3,4]. Despite the high incidence of CRC, it is one of the more curable malignancies if it is diagnosed in its early stages [5].

One challenge in diagnosing CRC is the possibility of a second, concurrent malignant lesion—a synchronous tumor—that is overlooked by sub-optimal diagnostic methods [6]. According to Warren and Gates, synchronous tumors are defined as two or more neoplasms in the same patient [7]. Diagnosis of a synchronous tumor must satisfy the following criteria: each tumor has to be malignant; each lesion has to be distinct; the likelihood that one tumor is the metastasis of the other must be excluded; and the synchronous lesions must be diagnosed at the same time or within six months from the first diagnosis [6,7]. The most pathologically advanced lesion is considered the index tumor, while the others are designated companion tumors [4].

The reported incidence of synchronous colorectal cancer (SCRC) ranges between 2–12% [4,8]. The development of SCRCs is presumed to be a stochastic or non-random process [9]. Exogenous factors such as diet, lifestyle, environment, and microbiome influence immune system cells and aberrant differentiation, and they can also contribute to the heterogeneity of the neoplastic process [10].

Surgical treatment is the primary therapeutic approach, followed by chemotherapy, which is mainly based on 5-Fluorouracil (5-FU) derivates [7,11,12]. In metastatic cases, targeted therapy focusing on the SCRCs with distinct molecular profiles requires molecular profile examination, knowing that BRAF and KRAS mutant tumors show resistance to anti-EGFR (epidermal growth factor receptor) therapy [13]. However, this has not yet become a therapeutic guideline.

After a review of the literature, the aim of our study was to propose better practices for the management of SCRCs, from diagnosis to molecular assessment of tumor heterogeneity.

## 2. Materials and Methods

### 2.1. Pre- and Intraoperative Assessments

This prospective study includes six individual patients diagnosed with SCRCs, each of them presenting with two carcinomas that benefited from surgical removal of the tumors between December 2018 and October 2020. The Ethical Approval of Clinical County Emergency Hospital of Targu-Mures, Romania, was obtained to include these patients in the prospective study. From each of the six patients, signed informed consent was obtained prior surgery for both permissions to perform the colectomy and the subsequent use of patient information in the publication of scientific data.

Preoperative diagnoses were based on colonoscopies and biopsies, except Case No. 6, where the patient was admitted to urgent care for abdominal pain and rectorrhagy, and a CT scan was the first to detect the synchronous lesions. In the cases with rectal tumors, MRI exams completed thoracic and abdominal CT-scans for preoperative staging (Figure 1, Figure 2, Figure 3, Figure 4 and Figure 5). For rectal cancer, a GE 1,5T MRI machine was used for pelvic MRI protocol (sequences used: Cor STIR, Ax T2—perpendicular on the tumor, Sag T2—parallel with the anal canal, Ax DWI, Ax T1). For the other tumors, thoraco-abdominal contrast-enhanced CT exams were performed using a Siemens Somaton 64 channels CT scanner with administration of intravenous iodinated contrast media (Optiray 350, Ioversol 74%, 350 mg I/mL) in a dose of 1 mL/kg of body weight with a flow rate ranging from 2 to 3 mL/s). In Case No. 3, due to technical issues, imaging data was not obtainable. Assessment of locoregional invasion was based on the following parameters: thickening of the intestinal wall, depth of tumor infiltration, and infiltration of the mesenteric fat (for rectal cancer). The diagnosis of an SCRC was based on the Warren and Gates criteria [7]. Additional colorectal lesions such polyps were also checked, along with LN status and the presence or absence of distant metastases. The imaging investigations sought to identify local and distant suspect LNs. The short axis diameter of each visible LN was registered. Malignancy suspicions were based on the size, shape, and heterogeneity of the examined LNs (Table 1).

Based on the imaging information adapted from Yamamoto et al. [14], with permission obtained from the authors, a map of abdominopelvic nodal stations was generated (Figure 1, Figure 2, Figure 3, Figure 4 and Figure 5).

All patients benefited from surgical excision of the SCRCs through total or partial colectomy with LN removal based on the imaged abdominopelvic map. All palpable locoregional LNs were prelevated according to current guidelines in order to correlate imaging and histopathological aspects. If suspect LNs were located outside the colectomy specimen anatomic area, the surgeon removed those LNs as well and sent them in separately for histopathological assessment.

### 2.2. Histopathological Assessment

Gross assessment of the surgical specimens was done according to the current guidelines and imaging map. All the palpable LNs were included for histological examination and comparison of imaging and microscopic features. Tumor deposits (TDs), number of LN metastases, and lymph node ratio (LNR) were included in the histopathological report for TNM staging.

### 2.3. Immunohistochemical and Molecular Examination

Immunohistochemical (IHC) stains were performed on formalin-fixed paraffin-embedded tissue (FFPE) blocks. Microsatellite status was checked based on the four IHC markers: MLH1 (clone ES05, Leica Microsystems GmbH; dilution 1:100, high retrieval solution—pH 10), MSH2 (clone 25D12, Leica Microsystems GmbH; dilution 1:50, high retrieval solution—pH 10), PMS2 (clone M0R4G, Leica Microsystems GmbH; dilution 1:50, high retrieval solution—pH 10), and MSH6 (clone PU29, Leica Microsystems GmbH; dilution 1:100, high retrieval solution—pH 10). Cases with nuclear positivity for all the four biomarkers were given microsatellite stable status (MSS).

For assessment of tumor molecular subtype, E-cadherin (clone NCH-38, Dako Agilent Technologies, Inc.; dilution 1:50, high retrieval solution—pH 10) and β-catenin (clone β-catenin-1, Dako Agilent Technologies, Inc.; dilution 1:150, high retrieval solution—pH 10) were used. For these markers, cellular localization was evaluated in both tumor cores and invasion fronts (tumor buds). Cases with membrane expression in both core and buds for E-cadherin and β-catenin were designated as epithelial subtypes; cases with loss of E-cadherin expression in the cell membrane with nuclear translocation of β-catenin were designated as mesenchymal subtypes.

To check the molecular status of the eight tumors, we extracted DNA from fresh tumor tissues using the Qiagen (Hilden-Germany) QIAmp DNA mini kit, according to manufacturer protocol. CE IVD kits (Quiagen therascreen KRAS and BRAF RGQ PCR kits) and the Rotor-Gene Q-MDx instrument from Qiagen were used for molecular analysis of KRAS and BRAF genes according to manufacturer protocols. In the case of KRAS genes, seven somatic mutations in codons 12 and 13 can be detected, while in the case of BRAF genes, five mutations in codon 600 of the gene were checked.

### 2.4. Follow-Up

One patient (Case No. 1) died three days after surgery. For the other cases, follow-up was performed between 6 and 18 months after surgery.

## 3. Results

### 3.1. Clinicopathological, Imaging and Histological Aspects

Cases 1 through 6 consisted of three males and three females with SCRCs, median age 70.16 years, and a total of 12 malignant tumors (two tumors per case). One patient (Case No. 2) declared that her sister had a rectal cancer, too. Additionally, in the same case, polyposis of the colon was found.

In two of the cases (Case Nos. 1 and 2), the preoperative CT-scan detected secondary tumors that were missed during endoscopy due to improper bowel preparation. In both cases, the tumors were in the cecum and were considered index tumors. In Case No. 6, the CT scan was the first performed and successfully detected both tumors and the right ovary metastasis. In one patient (Case No. 3), the imaging data was not available for pathological correlation, due to technical issues.

Five tumors were localized in the proximal colon, four in the distal colon, two in the rectum, and one at the level of the anal canal. All tumors were MSS ADKs (Figure 6), except the one from the anal canal (Case No. 4), which was an MSS squamous cell carcinoma (SCC) (Table 2). In Case No. 4, the index tumor was the ADK of the sigmoid colon, while the SCC with invasion limited to the muscularis propria was considered the companion tumor. Ten tumors were included in the epithelial subtype, with two (Case No. 4 and No. 5) hybrid subtypes: membrane expression for E-cadherin and β-catenin in the tumor core, with loss of E-cadherin and nuclear translocation of β-catenin in buds (Figure 7).

Regarding the LNs, only one of the cases (Case No. 2) did not show TDs or LN metastases (Table 2). Positive LNs were considered suspect during the imaging investigation (Case Nos. 4–6) and TD was identified preoperatively in Case No. 1 as a LN metastasis. The LNs that showed no suspicious imaging features correlated with negative LNs (Figure 1, Figure 2 and Figure 3).

Intraoperatively, one peritoneal metastasis (ileal) and a perirectal tumoral deposit (TD) were detected in Case No. 1 (Table 2).

### 3.2. KRAS and BRAF Gene Profiles

All tumors presented wildtype (non-mutated) BRAF gene status. For the KRAS gene, no mutations were identified in Case Nos. 3, 4 and 6. In three cases, we found KRAS mutations (Case Nos. 1, 2 and 5). In Case Nos. 1 and 2, there was discordance in the status between the lesions in each patient. In both patients, the index tumor presented KRAS mutations (12Val and 12Cys) and the companion one showed wildtype KRAS. In Case No. 5, both tumors showed codon 12 KRAS mutation (12Ala in the index tumor and 12Val in the companion tumor) (Table 3).

### 3.3. Postoperative Treatment and Follow-Up of the Patients

As the patient in Case No. 1 died and the patient in Case No. 2 was in an early stage, no chemotherapy was necessary for either. CT-scans and carcinoembryonic antigen (CEA) serum level monitoring showed that, in Case No. 2, there were no recurrences at 6-, 12-, and 18-month follow-ups.

In Case No. 3, the patient was diagnosed with stage IIIB adenocarcinomas of the ascending colon with MSS status; surgery was followed by adjuvant chemotherapy consisting of six 21-day cycles of intravenous oxaliplatin and oral capecitabine. The follow-up CT-scan showed no recurrence at 12 months post-surgery.

In Case No. 4, the index tumor that showed lymph node metastases was the adenocarcinoma of the sigmoid colon, and adjuvant chemotherapy was indicated using the same regimen as for Case No. 3. Due to patient’s underlying condition (anemia), COVID-era concerns, and other non-tumor-related comorbidities (hip surgery for coxarthrosis), chemotherapy was delayed, being initiated at 9 months after surgery, with no recurrence at 12-month follow-up.

In Case No. 5, both tumors having KRAS mutation, preoperatory radiotherapy (50 GY/23fr/42 days/pelvis) and oral capecitabine was initiated. No recurrence at 12-month follow-up.

Case No. 6 presented lymph nodes and systemic metastases being classified as stage IVB. Chemotherapy with oxaliplatin and oral capecitabine was indicated, same as cases No. 3 and 4. The patient is currently under chemotherapy.

## 4. Discussion

In the presented cases, the preoperative imaging assessment was done according to the Japanese Classification of Colorectal, Appendiceal, and Anal Carcinoma (JCCRC), developed and updated in 2019 by the Japanese Society of Cancer of the Colon and Rectum (JSCCR) guidelines [14]. This assessment proved extremely useful considering the strong concordance between imaging and histological assessments of the LNs.

Also, particularly valuable to the study was the complex evaluation of the cases by a transdisciplinary team using a variety of approaches including imaging and molecular assessment. We chose to evaluate the epithelial to mesenchymal transition (EMT), which is known to induce aggressive tumor behavior and higher rates of chemoresistance [14]. The hybrid phenotype, which is characterized by loss of E-cadherin and membrane-to-nuclear translocation of β-catenin in tumor buds [15] was observed in two of the cases.

Evaluating the MSS status was necessary for predictive purposes. It is known that over 10% of CRCs might present microsatellite instability (MSI) and do not benefit from 5-FU-based chemotherapy [12,13,16]. In this study, all of the cases were MSS tumors.

Examination of KRAS and BRAF mutation statuses is indicated in metastatic cases since wildtype KRAS and BRAF carcinomas are eligible for anti-EGFR therapy [13]. There are no guidelines regarding the discordant cases and their postoperative oncologic management [17,18,19,20]. Previous studies have reported that the presence of a mucinous component and BRAF mutation rate appear at a higher frequency in SCRCs than in solitary CRCs, but none of our cases had BRAF mutations. [13,19,20]. In the literature, the reported concordance rates for MSI, KRAS mutation, and BRAF mutation between tumors in the same patient varied from 9–30%, 11–40% and 14%, respectively [9,13,21]. In a study on 59 patients with SCRCs, Arakawa et al. found that, although low, there was significant correlation between the subtypes of MSI, BRAF mutation, and β-catenin between the lesions in each patient, while KRAS mutation had poor correlation [13]. Although one of our patients presented with a familial history of CRC the connection between genetic background and the incidence of SCRCs is not confirmed [17,18].

Although most of the SCRCs include two concomitant lesions, there have been reported cases with three, four, and even seven simultaneous tumors in the same patient [19,20]. Barium enemas and colonoscopies remain the primary modalities for detecting SCRCs [19,22]. While intra-operatory palpation of entire colon can help detect a concurrent CRC in cases of an acute onset of complete or partial obstruction, the sensitivity of SCRC diagnoses has a 50% rate of failure, especially in early staged tumors [6]. None of these methods are optimal, and tumors can be missed for various reasons, like poor bowel preparation and retained feces or inability to pass the endoscope through the lumen in cases of stenosing type tumors [6]. A complete colonoscopy implies visualization of the ileocecal valve and appendiceal orifice or terminal ileum [5].

Virtual colonoscopy and CT colonography are more accurate diagnostic methods, able to detect synchronous lesions in the colon proximal to an obstructing lesion; with a correct CRC staging rate of 81%, virtual colonoscopy and CT colonography demonstrate similar accuracy to conventional colonoscopies in identifying lesions greater than 1 cm [23]. Other useful methods are MRI colonography or CT colonography combined with PET [6]. Ultimately, a routine CT-scan could identify suspect concomitant lesions if they present with a narrowed intestinal lumen and visible pericolic LNs, as well as detect distant metastases. However, for preoperative staging of colorectal cancer, the gold standard remains the MRI [24].

Regarding LNs, size alone was not correlated with the presence of metastatic cells. Other morphological features, such as irregular borders or heterogenous signal intensity, display better sensitivity and specificity (85% and 97%, respectively) as indicators of positive LNs [25,26].

One issue that deserves attention is the presence of tumor deposits (TDs), which are usually identified under microscope by the pathologist. TDs are defined as nodules within the mesorectum that contain carcinoma cells without the obvious structure of a lymph node [27]. In one of our cases, TDs were observed on imaging exams and included in the group of suspect LNs (Case No. 1). This underscored the accuracy of CT-scan evaluations in such cases. Although LNs and deposits might be differentiated on MRI based on their relationship with vascular structures, confusion associated with the use of the CT-scan can easily be made [27].

The standard initial treatment for SCRCs is surgical resection. Some authors recommend a more extensive resection for patients with lesions in adjacent segments [11], while others suggest that a subtotal or a total colectomy should be performed even when synchronous lesions are in separate segments in order to reduce the possibility that other concomitant lesions may be overlooked leading to repeated surgery and an overall poorer prognosis [6]. One of our cases was treated with a total colectomy despite the tumors being situated in different segments of the bowel (caecum and rectum), but an association of colic polyposis may predispose the patient to new malignancies, making the more extensive procedure the better option. Of course, postoperative chemotherapy is still indicated for stages III and above [12,28,29].

In SCRCs with colon and rectal tumors, management becomes even more challenging, and decisions such as the extent of surgery or whether to initiate neoadjuvant RCT depend very much on the preoperative findings. None of our cases with rectal cancer were eligible for neoadjuvant RCT, and the stage I case was treated solely with a total colectomy with no sign of recurrence at 18 months.

A point of interest among these cases was the combination of an ADK in the sigmoid colon with a SCC of anal canal in Case No. 4. Anal carcinoma is a rare tumor, responsible for just 2–4% of reported incidents of lower digestive tract malignancies [29], and the synchronicity of this type of tumor appearing with a colorectal ADK is very rare [30,31]. This case underscored the importance of establishing the histologic type of concomitant tumors when planning the surgical resection and implementing postoperative RCT treatment [30].

The prognosis of patients with SCRCs is not well documented; various studies suggest the same, better, or worse outcomes than solitary CRC [9,20]. Life-long clinical follow-up is recommended, as local or distal recurrence is reported in 30–50% of the cases [32]. One of our patients, Case No. 1, died soon after the surgery due to metabolic imbalance, preventing us from following the progression of his disease and evaluating the efficacy of treatment decisions; however, no recurrence was noted in the other patients during their follow-up evaluations.

Our study had its limitations. Although the incidence of SCRCs is rising, it is still a rare pathology, explaining the small number of cases available for study. Additionally, due to lack of imaging studies, in Case No. 3 we failed to correlate the preoperative LNs status with the pathological findings. For more statistically significant results, further research for longer periods on larger cohorts of patients using a standardized preoperative evaluation scheme that includes colonoscopies and imaging studies in all cases and molecular analysis of each tumor is needed.

## 5. Conclusions

Due to the molecular heterogeneity of SCRCs, they should be carefully checked before any colorectal tumor surgery. CT and MRI scans can be successfully used to identify suspect LNs, but an in-house protocol should be standardized. This paper highlights, in the era of molecular pathology, the importance of molecular analysis in any SCRC cases.

## Figures and Tables

**Figure 1 diagnostics-11-00314-f001:**
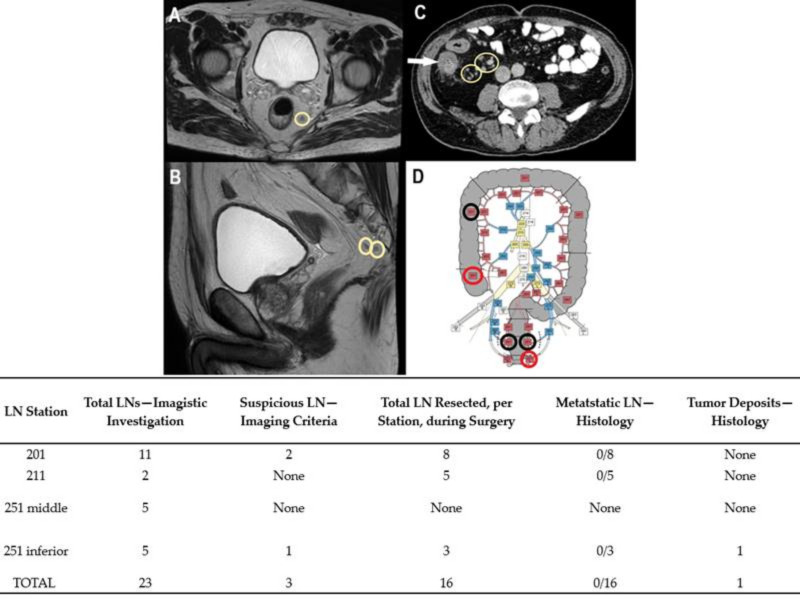
Case No. 1: synchronous cecal and rectal tumor. Circumferential thickening with inhomogenous enhancement and showing infiltration of the mesenteric fat at the level of the caecum (white arrow). Suspected lymph nodes (LNs) are visible on pelvic MRI scan, T2 weigthed FRFSE axial (**A**) and sagital (**B**) view. They are also marked on contrast-enhanced abdominal CT scan, venous phase, axial view (**C**). (**D**)—Based on imagistic aspects, the map of nodal stations is edited—adapted with permission from Yamamoto S et al. [14]—black circle—visible LN stations, without suspicious criteria; red circle- suspect LN stations. Table: imagistic-histopathological correlation of node stations. The two suspicious LN in 201 nodal station were negative by pathologic report. The perirectal tumor deposit was imagistically detected as suspected for LN metastasis.

**Figure 2 diagnostics-11-00314-f002:**
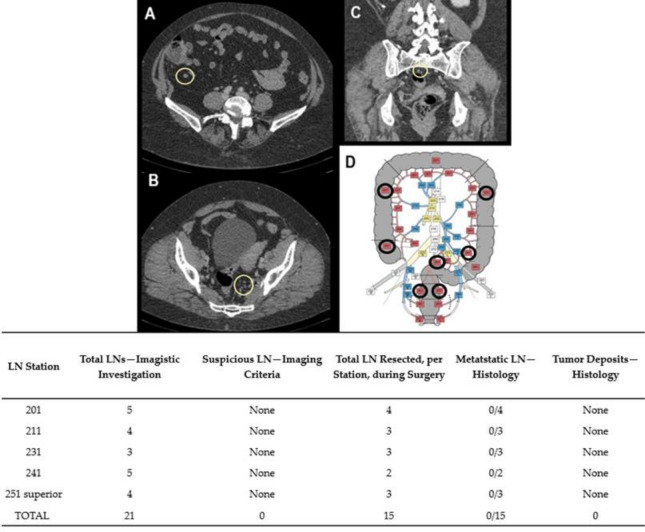
Case No. 2: synchronous cecal and rectal tumor. LNs can be seen on an abdominopelvic contrast-enhanced CT scan, venous phase, axial (**A**,**B**) and coronal view (**C**). No suspected LNs are detected. (**D**)—The map of LN stations—black circles emphasize LN stations, without suspicious features. Table: correlation of LN stations, based on imaging studies and pathological reports. Imaging did not detect any suspicious LNs that was confirmed by pathology report.

**Figure 3 diagnostics-11-00314-f003:**
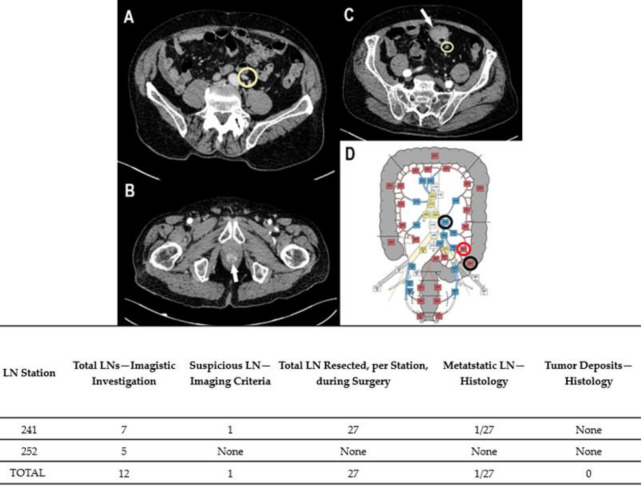
Case No. 4: synchronous sigmoid ADK and SCC of the anal canal. Circumferential thickening with inhomogeneous enhancement at the level of sigmoid, proves infiltration of the mesenteric fat (white arrow). Perisigmoid and inferior mesentery LN stations were seen on contrast-enhanced CT, venous phase, axial views (**A**,**C**). The tumor mass in the anal region shows heterogenous enhancement in the arterial phase (**B**), with no visible LNs. (**D**) A map of nodal stations involved was made: black circle—visible LN, without suspicious features; red circle—suspicious LN for malignancy. Table: map correlation showed that positive LN was successfully detected on CT scan.

**Figure 4 diagnostics-11-00314-f004:**
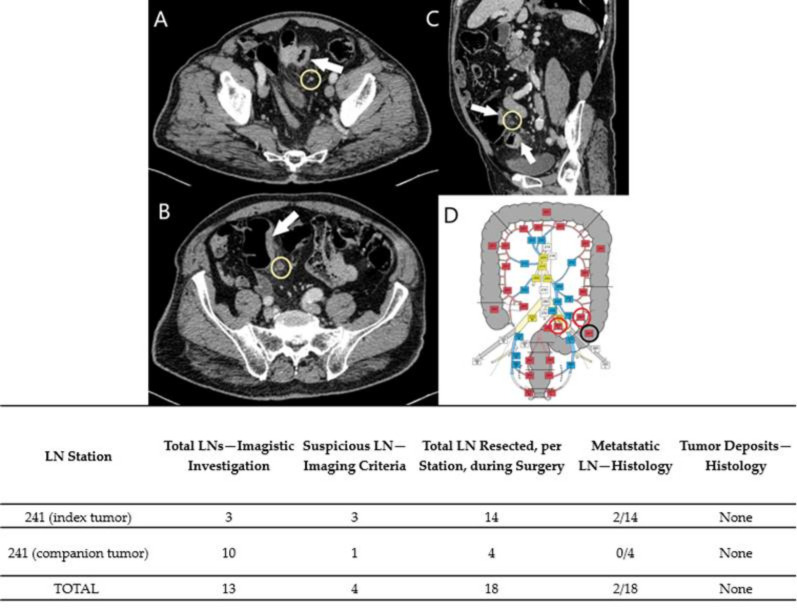
Case No. 5. Two synchronous sigmoid tumors. Parietal thickening of the colic wall with contrast-enhancement (white arrows), infiltration of pericolic fat. Association of regional LNs (yellow circle), some with suspicious criteria, seen on contrast-enhancement CT scan, venous-phase, axial-views (**A**,**B**) and sagittal view (**C**). A map of nodal stations was made: black circle: visible, non-suspicious LNs, red circle: suspicious LNs (**D**). Table: correlation of LNs stations - two of the four suspect LNs were positive, both of them found in the index tumor.

**Figure 5 diagnostics-11-00314-f005:**
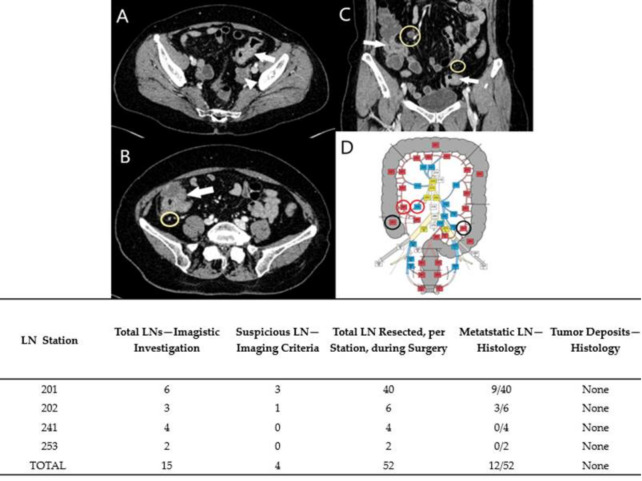
Case No. 6. Synchronous sigmoid and cecal tumors (white arrow), seen on contrast-enhanced CT scan, venous phase, axial views (**A**,**B**) and coronal view (**C**), with infiltration of pericolic fat, regional LNs (yellow circle) and left fallopian tube metastasis (white arrow-head). Map of nodal stations shows nodal stations with suspect LNs (rec circle) and nodal stations with visible LNs, without suspicious criteria (**D**). Table: correlation of suspect nodal stations with LN metastasis.

**Figure 6 diagnostics-11-00314-f006:**
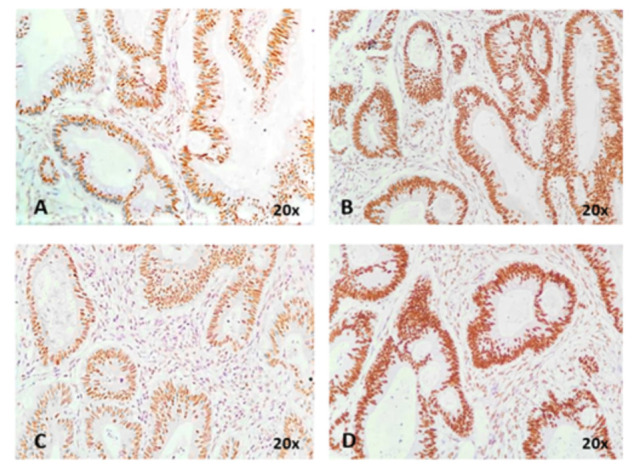
The microsatellite stable status is immunohistochemically proved by nuclear expression of MLH1 (**A**), MSH2 (**B**), PMS2 (**C**) and MSH6 (**D**).

**Figure 7 diagnostics-11-00314-f007:**
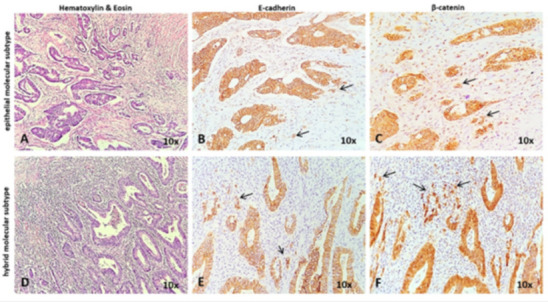
Histological aspect, correlated with the molecular subtype determined with E-cadherin and β-catenin. All tumors expressed membranar E-cadherin and β-catenin in the tumor core (**A**–**F**). In the epithelial subtype, the membrane expression of E-cadherin (**B**) and beta-catenin (**C**) can be seen in both tumor core and invasion area (buds). The hybrid molecular subtype is characterized by membranar expression of E-cadherin and beta-catenin in the tumor core (**E**,**F**) with nuclear positivity of β-catenin in the buds (**F**).

**Table 1 diagnostics-11-00314-t001:** Suspicious LNs based on imaging malignant features (size and morphologic characteristics).

Malignant Characteristics of LN	Roundness	Heterogeneity	Irregular Contour
<5 mm		All three characteristics	
5–10 mm		Association of any two characteristics	
>10 mm		Always suspicious	

**Table 2 diagnostics-11-00314-t002:** Clinicopathological characteristics of the patients.

	Case No. 1		Case No. 2		Case No. 3		Case No. 4		Case No. 5		Case No. 6	
**Sex**	**Male**		**Female**		**Female**		**Male**		**Male**		**Female**	
**Age (years)**	63		60		73		79		78		68	
	**I**	**C**	**I**	**C**	**I**	**C**	**I**	**C**	**I**	**C**	**I**	**C**
**Localization**	Cecal	Inferior rectum	Cecal	Superior rectum	Ascending colon	Ascending colon	Sigma	Anal canal	Sigma	Sigma	Cecal	Descending colon
**Method of detection**	CT	E	CT	E	E	E	E	E	E	E	CT	CT
**Histological type and grade**	G2-ADK	G2-ADK with mucinous component	G2-ADK with mucinous component	G2-ADK	G2-ADK with mucinous component	G2-ADK with mucinous component	G2-ADK	NK SCC	G2-ADK with mucinous component	Sigilocelular carcinoma	ADK, micropapillary variant	ADK, micropapillary variant
**Grade of tumor budding**	Low	Low	Low	Low	High	Moderate	Low	Low	High	Low	High	High
**No. Of positive LN/total LN resected**	0/16 and one perirectal tumor deposit		0/15		1/17 = 0.05		1/27 = 0.03		2/14 = 0.14		12/46 = 0.26	
**Distant metastasis**	Peritoneal metastasis		None		None		None		None		Right ovary and fallopian tubes bilateral	
**TNM staging**	pT3N1cM1a; pT3N1cM1a		pT2N0N0M0; pT1N0M0		pT4aN1M0; pT3N0M0		pT4N1M0; pT3N0M0		pT3N1bM0; pT4aN0M0		pT4bN2bM1; pT4bN0M1	
**AJCC Stage**	IVA		I		IIIB		IIIB		IIIB		IVB	
**Other colorectal lesions**	-		Colic polyposis		-		rectal polyp		sigmoidian polyps		-	
**Surgical procedure**	For index tumor—right hemicolectomy with ileotransverse anastomosis and ileal resection For companion tumor-rectal amputation		For both tumors- total colectomy with ileorectal anastomosis		For both tumors—right hemicolectomy		For sigmoid tumor -rectosigmoid ectomy (Hartmann I) For anorectal junction tumor—excision		For both tumors- left hemicolectomy		For both tumors—subtotal colectomy with segmentary ileal resection Histerectomy with bilateral anexectomy	
**Early postoperative complications**	Metabolical imbalance—death at three days after surgery		None		None		None		None		None	
**Chemotherapy/Radiotherapy**	-		Not indicated		Oxaliplatin perfusion + oral capecitabine		Oxaliplatin perfusion + oral capecitabine Delayed due to hip surgery and COVID period, initiated at 9 months after surgery		Preoperative radiotherapy (50 Gy/25fr/42 days/pelvis + oral capecitabine		Oxaliplatin perfusion + oral capecitabine	
**Follow-up**	-		No recurrence (18 months follow-up)		No recurrence (12 months follow-up)		No recurrence (at 12 months follow-up)		No recurrence (at 12 months follow-up)		Under chemotherapy	

I—index tumor, C—companion tumor, CT—computed tomography, E—endoscopy, ADK—adenocarcinoma, G2—moderately differentiated SCC—squamous cell carcinoma, NK—non-keratinized.

**Table 3 diagnostics-11-00314-t003:** Molecular subtyping, grade of tumor budding and status of KRAS and BRAF genes.

	Case 1		Case 2		Case 3		Case 4		Case 5		Case 6	
**Synchronous tumors**	I	C	I	C	I	C	I	C	I	C	I	C
**Molecular subtype**	E	E	E	E	E	E	H	E	H	E	E	E
**Grade of tumor budding**	G1	G1	G1	G1	G3	G2	G1	G1	G1	G3	G1	G1
**Microsatellitar status**	MSS	MSS	MSS	MSS	MSS	MSS	MSS	MSS	MSS	MSS	MSS	MSS
**KRAS gene (codon 12 and 13)**	***Mutated 12Val***	Wild-type	***Mutated 12Cys***	Wild-type	Wild-type	Wild-type	Wild-type	Wild-type	***Mutated 12Ala***	***Mutated 12Val***	Wild-type	Wild-type
**BRAF gene (V600E)**	Wild-type	Wild-type	Wild-type	Wild-type	Wild-type	Wild-type	Wild-type	Wild-type	Wild-type	Wild-type	Wild-type	Wild-type

I—index tumor, C—companion tumor; E—epithelial, H—hybrid. the bold and italic: the mutated cases.

## Data Availability

The database can be obtained from the authors, upon request.

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
