# Peer review of "Synchronous Colorectal Cancer: Improving Accuracy of Detection and Analyzing Molecular Heterogeneity—The Main Keys for Optimal Approach"

_diagnostics, 2021, doi:10.3390/diagnostics11020314_

Round 1

Reviewer 1 Report

The issue of the heterogeneity of synchronous and metachronous primary tumors is of great interest, expecially because nowadays the treatment of metastatic disease is driven by molecular profile, and by side.

However, the small number of cases (only four) and the short follow up time are severe limitations.

Issues to be clarified/discussed:

  1. As a general issue, some results are reported in the Discussion session, making difficult go through the manuscript

  1. The authors stress the role of imaging in the diagnosis of synchronous tumors of the large bowel, however, endoscopy remains the pivotal tool, but in case of left colon occlusive lesions. The authors should clarify side and stage of the two synchronous tumors diagnosed by imaging only and missed at colonoscopy, and if endoscopy was complete. Only in the discussion the reader knows that the colonoscopy was incomplete in two cases, but it is not specified which of the four cases

  1. The diagnosis of metastatic lymph-nodes by CT-scan or MRI is still matter of debate. The auhtors should clarify the statement in the last paragraph of the Material and Methods: it seems that only suspect lymph-nodes have been removed and not all the regional ones.

  1. The peritoneal metastasis (case 1) was diagnosed by imaging or during surgery?

  1. The authors should clarify the treatment and the local stage of all the 8 tumors:
    1. Case 1 = surgery for both right colon and rectal cancer; in table 2 only one TNM stage is reported
    2. Case 2 = did the patient undergo a total colectomy including both cecal and rectal cancer? in table 2 only one TNM stage is reported
    3. Case 3 had two synchronous primary tumors, both in the right colon? in table 2 only one TNM stage is reported
    4. Case 4 = how the anal cancer was treated?

  1. The presence of additional polips along the large bowel besides the invasive tumors should be described

Author Response

REVIEWER 1

Open Review

English language and style

( ) Extensive editing of English language and style required
( ) Moderate English changes required
(x) English language and style are fine/minor spell check required
( ) I don't feel qualified to judge about the English language and style

Yes

Can be improved

Must be improved

Not applicable

Does the introduction provide sufficient background and include all relevant references?

(x)

( )

( )

( )

Is the research design appropriate?

( )

( )

( )

(x)

Are the methods adequately described?

( )

( )

( )

(x)

Are the results clearly presented?

( )

( )

(x)

( )

Are the conclusions supported by the results?

(x)

( )

( )

( )

Comments and Suggestions for Authors

The issue of the heterogeneity of synchronous and metachronous primary tumors is of great interest, expecially because nowadays the treatment of metastatic disease is driven by molecular profile, and by side.

However, the small number of cases (only four) and the short follow up time are severe limitations. Two more cases were introduced – the number increased to a total of 6 cases

Issues to be clarified/discussed:

  1. As a general issue, some results are reported in the Discussion session, making difficult go through the manuscript – We have cancelled the un-necessary information. 
  1. The authors stress the role of imaging in the diagnosis of synchronous tumors of the large bowel, however, endoscopy remains the pivotal tool, but in case of left colon occlusive lesions. The authors should clarify side and stage of the two synchronous tumors diagnosed by imaging only and missed at colonoscopy, and if endoscopy was complete. Only in the discussion the reader knows that the colonoscopy was incomplete in two cases, but it is not specified which of the four cases – ”In two of the cases (Case Nos. 1 and 2), the preoperative CT-scan detected secondary tumours that were missed during endoscopy due to improper bowel preparation. In both cases, the tumors were located in the cecum and were considered index tumors” – stating the cases where CT detected the tumors, stating the location of tumors and the reason of missed lesions during endoscopy. We have added this information in  the main text

  1. The diagnosis of metastatic lymph-nodes by CT-scan or MRI is still matter of debate. The auhtors should clarify the statement in the last paragraph of the Material and Methods: it seems that only suspect lymph-nodes have been removed and not all the regional ones. - ”All patients benefited from surgical excision of the SCRCs through total or partial colectomy with LN removal based on the imaged abdominopelvic map. All palpable locoregional LNs were prelevated according to current guidelines in order to correlate imaging and histopathological aspects. If suspect LNs were located outside the colectomy specimen anatomic area, the surgeon removed those LNs as well and sent them in separately for histopathological assessment.” – explaining the protocol of excision of the LNs. We have added this information in  the main text

  1. The peritoneal metastasis (case 1) was diagnosed by imaging or during surgery? - Intraoperative, there was detected one peritoneal metastasis (ileal) and a perirectal tumoral deposit (TD) in Case No. 1 (Table 2). ” – the peritoneal metastasis was detected intraoperative.

  1. The authors should clarify the treatment and the local stage of all the 8 tumors:
    1. Case 1 = surgery for both right colon and rectal cancer; in table 2 only one TNM stage is reported – both TNMs and the type of surgery for each of the tumors mentioned in the table
    2. Case 2 = did the patient undergo a total colectomy including both cecal and rectal cancer? in table 2 only one TNM stage is reported – both TNMs and the type of surgery for both tumors (due to colic polyposis)
    3. Case 3 had two synchronous primary tumors, both in the right colon? in table 2 only one TNM stage is reported – Both TNMs mentioned
    4. Case 4 = how the anal cancer was treated? - Excision

  1. The presence of additional polips along the large bowel besides the invasive tumors should be described – described in the table.

Reviewer 2 Report

The topic of the study is current and beneficial for the issue of colorectal cancers treatment. 

Comments:

It is necessary to clearly define study design in abstract and methodology: ....prospective, analytic, observational, case-control???..

Abstract:

What is the article topic? Molecular heterogeneity in synchronous colorectal cancer or usefulness of CT? See in conclusions.

 "... the second tumour was missed at endoscopy but diagnosed by a subsequent computed-tomography-scan (CT-scan)..." This is not a information for abstract with presented title.

Article:

In methods will be useful to describe used standardized protocol of oncological staging before treatment. Imaging method?

The methodological protocol of documentation is beneficial, it was not observed at patient No.3. What is meant by: "...due to lack of imaging studies, in Case No. 3.." No imaging metod was performed in staging of case No.3?

The patient No.3 should be exclude when you would like to conclude:... CT and MRI scans can be successfully used to identify suspect LNs....

Please explain connection between: 1)the title "Molecular heterogeneity in synchronous colorectal cancer" and 2)in the conclusions "... CT and MRI scans can be successfully used to identify suspect LNs, but an in-house protocol should be standardized..."

The role of molecular testing in differential diagnoses is clear but its impact to usefulness of imaging methods?

You have: to explain connection X to solve it in separate articles X to change the title and aim decription. 

Author Response

REVIEWER 2

Open Review
(x) I would not like to sign my review report
( ) I would like to sign my review report
English language and style
( ) Extensive editing of English language and style required
( ) Moderate English changes required
( ) English language and style are fine/minor spell check required
(x) I don't feel qualified to judge about the English language and style

    Yes    Can be improved    Must be improved    Not applicable
Does the introduction provide sufficient background and include all relevant references?    ( )    (x)    ( )    ( )
Is the research design appropriate?    ( )    (x)    ( )    ( )
Are the methods adequately described?    ( )    ( )    (x)    ( )
Are the results clearly presented?    ( )    (x)    ( )    ( )
Are the conclusions supported by the results?    ( )    ( )    ( )    (x)
Comments and Suggestions for Authors
The topic of the study is current and beneficial for the issue of colorectal cancers treatment. 
Comments:
It is necessary to clearly define study design in abstract and methodology: ....prospective, analytic, observational, case-control??? ”This prospective study includes six individual patients diagnosed with SCRCs, each of them presenting with two carcinomas that benefited from surgical removal” We have added this information in  the main text
Abstract:
What is the article topic? Molecular heterogeneity in synchronous colorectal cancer or usefulness of CT? See in conclusions. New title of the article: ”Synchronous colorectal cancer: improving accuracy of detection and analyzing molecular heterogeneity – the main keys for optimal approach”
 "... the second tumour was missed at endoscopy but diagnosed by a subsequent computed-tomography-scan (CT-scan)..." This is not a information for abstract with presented title. Changed title, showcasing the importance of detecting synchronous tumors with imaging methods

Article:
In methods will be useful to describe used standardized protocol of oncological staging before treatment. Imaging method? ”In the cases with rectal tumors, MRI exams completed thoracic and abdominal CT-scans for preoperative staging”
The methodological protocol of documentation is beneficial, it was not observed at patient No.3. What is meant by: "...due to lack of imaging studies, in Case No. 3.." No imaging metod was performed in staging of case No.3? – ”In case No. 3, due to technical issues, imaging data was not obtainable.”
The patient No.3 should be exclude when you would like to conclude:... CT and MRI scans can be successfully used to identify suspect LNs....
Please explain connection between: 1)the title "Molecular heterogeneity in synchronous colorectal cancer" and 2)in the conclusions "... CT and MRI scans can be successfully used to identify suspect LNs, but an in-house protocol should be standardized..." – Title changed, showing the importance of detecting a possible concomitant lesion with CT and/or MRI and also to analyze the molecular and genetic aspects for a targeted treatment.
The role of molecular testing in differential diagnoses is clear but its impact to usefulness of imaging methods? – A synchronous tumor can be easily missed if the endoscopy has suboptimal bowel preparation and a CT scan is very useful, not only to characterize the primary tumor, but also detect a possible second lesion. Without a proper imaging management we cannot have a proper molecular view of the synchronous tumors.
You have: to explain connection X to solve it in separate articles X to change the title and aim decription. – Changed the title of the articol and the aim, showing the importance of a multidisciplinary approach, from detection and staging with the help of imaging methods to analyzing the molecular aspects of tumors, for a better management. This paper highlights the role of ONCOTEAM in such challenging cases.

Round 2

Reviewer 2 Report

The version of article with six cases and a better focused title (Synchronous colorectal cancer: improving accuracy of detection and analyzing molecular heterogeneity – the main keys for optimal approach) is acceptable. Congratulations.